# Synthesis and Investigation of Properties of Beryllium Ceramics Modified with Titanium Dioxide Nanoparticles

**DOI:** 10.3390/ma16196507

**Published:** 2023-09-30

**Authors:** Alexandr Pavlov, Zhuldyz Sagdoldina, Almira Zhilkashinova, Nurtoleu Magazov, Zhangabay Turar, Sergey Gert

**Affiliations:** 1Surface Engineering and Tribology Research Center, Sarsen Amanzholov East-Kazakhstan University, Ust-Kamenogorsk 070000, Kazakhstan; 2National Laboratory of Collective Use, Sarsen Amanzholov East-Kazakhstan University, Ust-Kamenogorsk 070000, Kazakhstan; 3Center of Excellence “VERITAS”, Daulet Serikbayev East Kazakhstan Technical University, Ust-Kamenogorsk 070000, Kazakhstan

**Keywords:** beryllium oxide, TiO_2_ nanoparticles, microstructure, apparent density, microhardness, microwave radiation, reflection losses

## Abstract

Samples of beryllium ceramics, with the addition of micro- and nanoparticles TiO_2,_ have been obtained by the method of thermoplastic slip casting. The microstructure of batch ceramics, consisting of micropowders and ceramics with TiO_2_ nanoparticles sintered at an elevated temperature, has been investigated. It was found that the introduction of TiO_2_ nanoparticles leads to changes in the mechanisms of mass transfer and microstructure formation, and the mobility of TiO_2_ at interfacial grain boundaries increases, which leads to the formation of elements of a zonal shell structure. The reduction of intergranular boundaries leads to an increase in density, hardness, and mechanical strength of ceramics. The whole complex of properties of the synthesized material, with the addition of TiO_2_ nanoparticles in the amount of 1.0–1.5 wt.%, leads to an increase in the ability to absorb electromagnetic radiation in the frequency range of electric current 8.2–12.4 GHz. The analysis and updating of knowledge on synthesis, and the investigation of properties of beryllium ceramics modified by nanoparticles, seems to be significant. The obtained results can be used in the creation of absorbers of scattered microwave radiation based on (BeO + TiO_2_) ceramics.

## 1. Introduction

Presently, vacuum-dense ceramic materials that weaken microwave radiation in a wide frequency range are of particular interest in the development of modern electronic devices [1]. Such materials have the ability to absorb electromagnetic waves in certain frequency ranges and convert them into thermal energy [2]. The rapid development of vacuum electronic devices operating in extreme environments [3] requires microwave attenuation materials to have high thermal conductivity, fine grain structure, and mechanical strength, in addition to excellent microwave absorption capability. All these are necessary to prevent damage to electronic devices operating in corrosive environments [4]. Of all the known composite materials that meet the stated requirements, beryllium oxide [5] is the most promising due to its outstanding characteristics such as high thermal conductivity, good mechanical properties, high temperature stability, and moderate dielectric constant [6]. The unique combination of physical and chemical properties of beryllium oxide [7] determines a wide range of its use in various fields of modern engineering and special instrumentation [8]. High radiation resistance of BeO ceramics [9], thermal conductivity, dielectric strength, and transparency to X-ray, UV, IR and microwave radiation make BeO ceramics a promising material for use in a variety of devices and instruments of electronic technology for critical applications [10]. Pure BeO ceramics has high thermal conductivity, about 280–320 W/(m·K) and increased electrical resistance (~1 × 10^15^ Ohm·cm) [11], comparable to the thermal conductivity of chemically pure copper (~400 W/(m·K)) [12].

Various additives in the composition of BeO ceramics can lead to the presence of developed interfaces and intergranular interactions that affect the physicochemical and performance characteristics of composite ceramics. Titanium dioxide is one of such additives that can significantly change the conductive and other properties of BeO ceramics used as bulk electromagnetic energy absorbers [13].

The available data on the BeO–TiO_2_ system are contradictory, and the question of the existence of any beryllium titanates remains open. Thus, according to the authors [14], the results of the study of electrically conductive ceramics from the BeO:TiO_2_ mixture show that the two oxides involved create independent substructures in the mixed ceramics. The TiO_2_ substructure provides the electrical conductivity of the composite ceramic. The system is free of compounds; a single eutectic exists at a TiO_2_ content of about 85 wt.% and a temperature of 1670 °C. Currently, BeO is not known to form any “titanates” when interacting with TiO_2_ [15]. However, in paper [16] using the density functional theory, the structural, thermodynamic, and optoelectronic behavior of BeTiO_3_ perovskite in the pressure range 0–160 GPa was modeled. It was found that the studied perovskite is thermodynamically stabilized in the cubic phase, along with structural and mechanical properties.

It is known that the addition of TiO_2_ microparticles to the composition of BeO-based ceramics, with appropriate heat treatment in a weakly reducing environment of carbon monoxide, leads to the appearance of electrically conductive and dielectric properties of ceramics [17]. The main advantage of (BeO + TiO_2_) ceramics is the thermodynamic stability of properties, high mechanical strength, thermal conductivity, and excellent absorption characteristics [18]. It was found that the addition of 30 wt.% TiO_2_ to BeO ceramics leads to a significant increase in the dielectric constant and the degree of TiO_2_ reduction, which is accompanied by an increase in the dielectric loss angle tangent [19].

It was first reported in [20] about obtaining the material of BeO + TiO_2_ composition. It is known that the addition of TiO_2_ to BeO ceramics after sintering, under the same thermodynamic conditions in the amount from 5 to 40 wt.%, is accompanied by an increase in density, dielectric constant, and other properties [21].

At present, the composition BeO + 30 wt.% TiO_2_ (μm) is a commercially available microwave ceramic under the brand name BT-30.

The attenuation coefficient of microwave radiation passing through (BeO + TiO_2_) ceramics in the frequency range of 8.5–12 GHz is 19–20 dB. Titanium in relation to BeO has relatively low solubility 10^−3^–10^−4^ at. % and forms many oxides: TiO_2_, Ti_2_O_3_, Ti_3_O_5_ and others [22].

When such ceramics are heated in the temperature range from 750 to 1000 °C in hydrogen, TiO_2_ is transformed into the compound Ti_2_O_3_. Under the action of temperature 2000 °C in hydrogen medium under pressure 13–15 MPa, TiO_2_ is reduced to TiO. In the reducing environment of carbon monoxide under the action of temperature, the reaction TiO_2_ + C → TiO + CO takes place [23]. During thermal treatment of (BeO + TiO_2_) ceramics in the atmosphere of carbon monoxide, carbon penetrates into its internal structure [24]. In turn, in the process of sintering (BeO + TiO_2_) ceramics in the atmosphere of carbon monoxide, carbon residues in the billets, carbon from the graphite heater, and graphite grits, reducing gas CO, and interacting with TiO_2_ leads to the replacement of part of the oxygen vacancies in the anionic sublattice of TiO_2_ [25]. The process of isomorphic substitution of titanium ions and introduction of carbon ions into the inter-nodes of the TiO_2_ crystal lattice is also observed [26].

Therefore, the electrical conductivity of (BeO + TiO_2_) ceramics is primarily affected by the carbon impurity. Thus, if there will be insufficient oxygen in the furnace atmosphere during sintering of (BeO + TiO_2_) ceramics, the reactions of its substitution by carbon ions are preferable and, consequently, the formation of oxygen vacancies in the structure of TiO_2_. In the case of an excess of oxygen in the furnace atmosphere during sintering, the most probable reactions are the introduction of carbon into the inter-nodes and the substitution of titanium by carbon ions.

Thus, in the process of TiO_2_ reduction, when the oxidation degree changes in the series Ti^4+^ → Ti^3+^ → Ti^2+^, the number of d-electrons increases. The reduction leads to partial occupancy of a narrow, weakly conducting 3d-zone, which is vacant in stoichiometric TiO_2_ [27]. The partially reduced TiO_2_ is a semiconductor with an electronic type of conductivity [28]. According to the studies carried out in [29], it was found that oxygen ion vacancies are the defects of rutile, and in [30] it is shown that it is titanium ions that are the defects of rutile, with TiO_2_ ↔ Ti^n+^ + ne− + O_2_. It is assumed that titanium ions are located in the interstitials of the crystal lattice. The calculation of the electronic structure of TiO_2_ shows that instead of Ti and O ions, the introduction of carbon is energetically favorable [31]. Thus, the possibility to regulate the process of carbon introduction allows for control of the values of electrical conductivity of (BeO + TiO_2_) ceramics and to influence its ability to absorb electromagnetic radiation [32].

The authors of the present study found that the improvement of the electrophysical properties of such ceramics can be achieved by introducing TiO_2_ nanoparticles into its composition [33], which will contribute to the expansion of the operating frequency range, increase the stability of parameters during operation and exposure to external factors, and expand the technical characteristics in the field of special instrumentation. The authors also found that when the content of nanoparticles increases above 2.0 wt.% the viscosity of the slurry increases significantly, and it is not possible to mold the workpiece by the slurry casting method.

The main properties of TiO_2_ nanoparticles that contribute to the structural and electrophysical properties of such ceramics include the fact that with a decrease in the size of TiO_2_ particles, the absorption of electromagnetic energy occurs with the entire volume of the particle [34]. The physical processes occur during the sintering of ceramics based on BeO with the content of micro- and nanoparticles TiO_2_. New regularities of mass transfer and microstructure formation during sintering of the system (BeO + TiO_2_) modified by TiO_2_ nanoparticles emerge. This occurs due to an increase in the diffusion mobility of TiO_2_ at the interfacial boundaries of microcrystals, which leads to a decrease in porosity and an increase in the physical and mechanical properties of ceramics. Nanocrystalline addition leads to an increase in the number of active centers, bulk and surface defects available for reactions. The reduction reactions proceed more efficiently and the electrical and chemical properties of the synthesized material are changed [35]. At present, the current production of beryllium oxide ceramics is established in the USA, China and Kazakhstan, so the analysis and updating of knowledge on the study of the properties of nanostructured ceramics using this strategically important material seems to be very significant. Thus, the goal of this study is to obtain prototypes of commercially available ceramics, composition: BeO + 30 wt.% TiO_2_ with the addition of nanoparticles of titanium dioxide to study the structure, physical and mechanical properties and the ability of the synthesized material to absorb electromagnetic radiation. Maintaining the mass ratio of components will make it possible to trace the change in the properties of ceramics in comparison with the serial sample, which meets the requirements of the standards of the radioelectronic industry.

## 2. Materials and Methods

Experimental samples were prepared on the basis of micropowders of beryllium oxide with purity 99.9 wt.%, titanium dioxide with purity 99.5 wt.%, and nanopowders of titanium dioxide. The method of obtaining this is based on low temperature combustion of purified TiCl_4_ in the vapor phase in the presence of a catalyst. In this method of preparation, the morphology of the nanoparticles is crystals of a predominantly elongated shape, with a size of 50–80 nm, a diameter of 10–15 nm, and purity 99.9 wt.%. The particle sizes and surface morphology of the powders used are shown in Figure 1.

Introduction of nanoparticles into the micron matrix of BeO and TiO_2_ powders was carried out using an impeller-type reactor, the working body of which was filled with a liquid electrically nonconductive substance that does not react chemically with the particle material. Micro- and finely dispersed powders were loaded in the ratio of dry fraction to liquid fraction 1:7. The device of the reactor allows mixing of the charge components in the horizontal direction due to the rotation of blades, and the movement of flows in the liquid in the vertical direction occurs under the action of compressed air pressure supplied to the bottom of the reactor, providing effective distribution of TiO_2_ nanoparticles throughout the volume of the charge. 

Further, a thermoplastic slurry based on a wax–paraffin bond was prepared. The billet was molded on a casting machine for molding thermoplastic slurries into a collapsible mold. Firing of the binder from the billet was carried out in a muffle furnace at 1200 °C in aluminum oxide powder backfill. Sintering of the billet was carried out in a vacuum furnace with a graphite heater, packing the billet in a crucible made of beryllium oxide. The inside of the crucible was lined with molybdenum sheet. After laying the billets, the crucible was closed with a beryllium lid with technological holes for saturation of the billet with reducing gas CO. Sintering of the billet was carried out at 1660 °C with a holding time of 1 h. All experimental samples were obtained under the same regime, and saturation of the billet with reducing gas CO for all samples was made under the same conditions.

Thus, a series of experimental samples were obtained with the addition of TiO_2_ nanoparticles according to Table 1.

The sample of P0 composition is a serial sample of absorbing ceramics without the addition of nanoparticles. 

Microstructure, particle size distribution and phase analysis of powders and sintered samples were studied on a scanning electron microscope JSM-6390LV, 2007 (JEOL Ltd., Tokyo, Japan), with a resolution in a high vacuum up to 3 nm and the possibility of imaging in secondary and reflected electrons. The magnification of the microscope ranged from 5× to 300,000× at accelerating voltages from 0.5 to 30 kV. Some photographs of the microstructure were also obtained on a Hitachi SU3500 scanning electron microscope (Hitachi, Tokyo, Japan). The magnification of the microscope is from 5× to 300,000× at accelerating voltages from 0.3 to 30 kV.

The study of phase composition of synthesized ceramics was carried out by X-ray diffraction and implemented using a powder X-ray diffractometer D8 Advance ECO (Bruker, Berlin, Germany). The diffractograms were taken in Bragg–Brentano geometry in the angular range 2θ = 20–90°, with a step of 0.05° and a spectral acquisition time of 1.5 s at the point. To analyze the obtained diffractograms, as well as to determine the phase composition and structural parameters, we used the software DiffracEVA v. 4.2. The PDF-2 database was used to refine the phases.

Analyses to determine the values of apparent density were carried out according to the method described in GOST 2409-95 “Refractories. Method of determination of apparent density, open, total and closed porosity, water absorption”.

Determination of microhardness of the samples was carried out using the indentation method with the Vickers method. Diamond pyramid was used as an indenter, pressure force—500 N.

Mechanical strength was measured according to GOST 24409-80 “Ceramic electrotechnical materials”.

The particle size distribution and other studies of the ceramic microstructure were carried out using a BX-51 optical microscope (Olympus Corporation, Tokyo, Japan) with a digital camera and SIAMS 800 image analysis software.

Reflection coefficient values (S11/S22 Lin Mag) were measured using Keysight Technologies “Vector network analyzer P9373A” (Keysight Technologies, Santa Rosa, CA, USA) in the frequency range of 8.2–12.4 GHz.

To obtain reliable data, at least ten samples from each batch were measured. If the sample had defects in the form of cracks, shells and pores, such a sample was excluded from the experiment.

## 3. Results and Discussion

Beryllium oxide in relation to TiO_2_ is an inert compound, i.e., there is no chemical interaction potential between them. It is known that the solubility of Ti in BeO varies at the level of 10^−3^–10^−4^ at.%. The system TiO_2_–BeO belongs to simple eutectic. In the literature there is information only about the presence of a region of solid solutions extending from 85 to 100 wt.% TiO_2_, formed during melting in an oxygen–hydrogen flame [36].

On the phase diagram of TiO_2_–BeO [36] above the temperature of 1670 °C, titanium dioxide passes into the liquid phase. Therefore, in order not to “lose” mechanical properties due to crystal growth, the sintering temperature of (BeO + TiO_2_) ceramics should not be higher than 1670 °C. Depending on the sintering mechanism, the microstructure of ceramics is also determined. In the case of the solid-phase sintering process, grains are formed by the processes of pore dissolution and growth of individual crystals.

The data of entropy calculation of equilibrium constants allow us to identify four main chemical reactions that can occur at a temperature of 1660 °C, as shown in Table 2.

According to the results of the calculation of possible electrically conductive phases responsible for the conductivity of ceramics composition (BeO + $TiO_{2}^{\mu m}$ + $TiO_{2}^{nano}$) sintered at T = 1660 °C, the main electrically conductive phases are TiO, Ti_2_O_3_, Ti_3_O_5,_ and TiH. It is known that the compound Ti_2_O_3_ when fused with TiO_2_ under the influence of temperature (according to different data, at 1600–2000 °C) forms a phase based on Ti_3_O_5_, which is a double oxide of tri- and tetravalent titanium: $Ti_{2}^{III}O_{3}+Ti^{IV}O_{2}\to Ti_{3}^{10/3}O_{5}$.

It is a redox reaction: $Ti^{IV}+2/3e^{-}\to Ti^{10/3}$ (reduction) and $2Ti^{III}-2/3e^{-}\to 2Ti^{10/3}$ (oxidation), that is, TiO_2_ is an oxidizing agent and Ti_2_O_3_ is a reducing agent. Thus, the increase in sintering temperature and chemical activity of TiO_2_ nanoparticles leads to an increase in the percentage of the Ti_3_O_5_ phase in the ceramic structure. The effect of conductivity and electromagnetic wave absorption is achieved by adjusting the properties of ceramics by thermodiffusion of ions of variable valence into it, and creating a second phase with increased conductivity. By quantitatively changing the ratio of BeO and titanium-containing phases, as well as by regulating the degree of non-stoichiometry of titanium oxides, when the oxidation degree changes in the series Ti^4+^ → Ti^3+^ → Ti^2+^, it is possible to effectively control the properties of such ceramics. 

Commercially produced absorption ceramics has a composition of 70 wt.% BeO + 30 wt.% TiO_2_, sintered at a temperature of 1520–1530 °C (sample P0). The results of the study of the microstructure of ceramics made of BeO–TiO_2_ micropowders indicate that such ceramics is a two-phase mechanical mixture, where the white color represents the TiO_2_ phase and the dark color represents the BeO phase, as shown in Figure 2a,b.

The temperature of solid-phase sintering of pure BeO ceramics in vacuum is 1920–1950 °C. Such ceramics has a density of 2.5–2.9 g/cm^3^; the average grain size is 20–40 μm [37] and has a dense structure with uniformly distributed grains, as shown in Figure 3a.

In the microstructure of (BeO + TiO_2_) ceramics made of P0 micropowders, as shown in Figure 3b at ×900 magnification, a rather porous and friable structure of the BeO phase is observed, which indicates that the temperature is not sufficient for the beginning of solid-phase sintering processes. In solid-phase sintering, the free energy changes with the change in the surface area of the crystal, and with the replacement of the solid–gas interface with the solid–solid interface [38]. Increasing the sintering temperature of serial ceramics consisting only of micropowders leads to the fact that the sample swells and loses density which affects geometric parameters, and the growth of TiO_2_ grains occurs. 

The introduction of TiO_2_ nanoparticles in the composition of (BeO + TiO_2_) ceramics in the range from 0.5 to 1.5 wt.%, (Table 1) allows the sintering temperature of ceramics to increase by 130 °C. Thus, nanoparticles in the composition of ceramics contribute to restraining the growth of micron crystals of BeO and TiO_2_ under the action of surface and bulk diffusion processes. The microstructure of such ceramics is presented in Figure 4.

Ceramics with the addition of nanoparticles has a denser structure; the intergranular boundaries between TiO_2_ and BeO are reduced, and TiO_2_ grains have an irregular shape as they penetrate into the intergranular spaces of BeO through better wetting and shrinkage of ceramics during sintering. The introduction of TiO_2_ nanoparticles changes the mechanisms of mass transfer and microstructure formation. Changes in the mechanisms of surface diffusion and the mobility of TiO_2_ across the interfacial grain boundaries lead to the formation of zonal shell structure elements, as shown in Figure 5a–c.

Thus, electron microscopy revealed the existence of TiO_2_ solid phase and liquid solution spreading at a certain temperature on the surface of BeO microcrystals, likely representing a mixture of oxides as components of the solution Ti_4_O_7_ + TiO + Ti_3_O_5_ + Ti_2_O_3_. Thus, the condensed medium formed by heating TiO_2_ in a reducing atmosphere is not only the initial TiO_2_ oxide in solid or liquid states.

The results of the chemical microanalysis interpretation of the shell structure fragment, as shown in Figure 5b (highlighted with a circle), are presented in Figure 6.

As it is known [39] to create capacitor segmental ceramics characterized by increased temperature stability ε, mixtures of two phases of barium titanate are used. The second phase is a solid solution of heterovalent substitution of barium titanate with additives of simple or complex oxides. The grains of such ceramics have a zonal shell structure: each grain consists of a central part, barium titanate, and a shell–solid solution. The fact that the heterogeneous system is realized within a single grain ensures small changes in ε over a wide temperature range.

The combination of these factors leads to the formation and development of interparticle boundaries and, consequently, to an increase in the density, hardness, and mechanical strength of the synthesized material, as shown in Table 3.

TiO_2_ nanoparticles in the sintering process of (BeO + TiO_2_) ceramics promote the formation of Ti_3_O_5_ phase due to the increased chemical activity and decrease in the interaction temperature during the reaction Ti_2_O_3_ + TiO_2_ → Ti_3_O_5_. In turn, the presence of TiO_2_ nanoparticles can increase the sintering temperature of ceramics. The growth of micron crystals of BeO and TiO_2_ is restrained under the action of mass transfer, surface and bulk diffusion processes. As a result, it was possible to obtain samples with increased density, while maintaining a homogeneous fine-grained structure with a 1–5 µm grain size of TiO_2_.

The main physical and mechanical characteristics of ceramics consisting of micropowders and ceramics modified with TiO_2_ nanoparticles are shown in Table 4.

In addition, TiO_2_ nanoparticles have a strong influence on crystallization processes, changing the rate of formation and growth of new phases, size distribution, and shape of crystals. The high value of specific surface area of powders consisting of TiO_2_ nanoparticles can increase the diffusion rate and the rate of reduction reactions.

Next, the reflection coefficient (S11/S22 Lin Mag), the ratio of the complex amplitude of the reflected wave current to the complex amplitude of the incident wave current, was investigated on the “Vector network analyzer P9373A” in the frequency range of 8.2–12.4 GHz. The results are shown in Figure 7.

As shown in Figure 7, the reflection coefficient of ceramics without TiO_2_ nanoparticles addition at 8.2 GHz is the value of 0.88, then there is a monotonic sinusoidal decrease in the coefficient with the increase of electric current frequency. The first peak of 0.86 is observed at 9.3 GHz, the next peak of 0.84 at 10.3 GHz, then 0.82 at 11.4 GHz. At the introduction of TiO_2_ nanoparticles in the amount of 0.5 wt.%, the curve of dependence of the reflection coefficient on the frequency of electric current practically matches the curve for the P0 sample. It should be noted that in the microstructure of sample P1, only single fragments of zonal shell structure are observed, which apparently makes the greatest contribution to the loss of electromagnetic radiation. At the content of TiO_2_ nanoparticles—1.0 wt.% there is a sharp jump in the values of the reflection coefficient from 0.85 to 0.95 at a frequency of 8.2 GHz. With the increasing frequency of the electric current, the reflection coefficient also changes along a sinusoid with increasing amplitude. At the content of TiO_2_ particles—1.5 wt.% there is not a large increase in peak values, and the amplitude of reflection coefficient oscillation decreases.

The occurring changes in adsorption properties can be explained by the fact that titanium ions in the process of heat treatment of (BeO + TiO_2_)-ceramics are in bi-, tri-, and quadrivalent states. This promotes the occurrence of electron exchange between the localized states of Ti^2+^ ↔ Ti^3+^ ↔ Ti^4+^ ions. As is already known, in the Ti–O system the main condensed phases are five oxides, the most refractory of which is TiO_2_ with a melting point of 1912 °C. For the other oxides it is 1837 °C (Ti_2_O_3_), 1777 °C (Ti_3_O_5_), 1757 °C (TiO), and 1687 °C (Ti_4_O_7_). During sintering at 1660 °C in (BeO + TiO_2_) ceramics with the addition of TiO_2_ nanoparticles, the melting point of refractory oxides decreases. The obtained diffractograms of the studied samples, as shown in Figure 8, indicate a high degree of structural ordering and a more efficient transformation of the crystalline structure of TiO_2_ compound into an electrically conductive Ti_3_O_5_ compound.

The main contribution, according to the phase analysis in the structure of ceramics, corresponds to the phases of titanium dioxide (rutile) and beryllium oxide. Additionally, in the structure there are impurity inclusions characteristic for tetragonal TiH_2_ and orthorhombic Ti_3_O_5_ phases (Table 5), the content of which varies depending on the amount of introduced nano additives. The deviation of lattice parameters is associated with deformation processes occurring as a result of the formation of ceramics, as well as the presence of impurity phases and solid solutions of substitution and introduction.

Thus, the main chemical transformations occurring under conditions of high temperature synthesis of (BeO + TiO_2_) ceramics have been established, with the addition of TiO_2_ nanoparticles during heating in a reducing atmosphere. The X-ray phase method shows that in the samples of (BeO + TiO_2_) ceramics—along with reflexes from the crystalline structure of BeO, diffraction peaks of TiO_2_ corresponding to the crystalline modifications of rutile and anatase, as well as the background in the region of small angles—some amounts of the amorphous phase are fixed. In addition, weak reflections that are likely corresponding to titanium carbide and other phases, but which could not be identified, were recorded.

The degree of TiO_2_ reduction in BeO ceramics is related to the magnitude of dielectric losses. Deviation of TiO_2_ from the stoichiometric composition leads to a strong change in its electrophysical properties. According to the results of the authors’ studies [33,34,35], the main absorbing phase of microwave radiation is the semiconducting non-stoichiometric compound Ti_3_O_5_, formed by the reduction of TiO_2_ dioxide during the thermal treatment of ceramics in a reducing medium.

## 4. Conclusions

The main phases formed during sintering of (BeO + TiO_2_) ceramics in reducing medium—TiO, Ti_2_O_3_, Ti_3_O_5_ and TiH—have been identified by the method of entropic calculation of equilibrium constants of chemical reactions. Introduction of TiO_2_ nanoparticles in the composition of ceramics 70 wt.% BeO + 30 wt.% TiO_2_ in the amount of 0.5–1.5 wt.% allows for the increase of the sintering temperature of such ceramics from 1530 to 1660 °C, which leads to changes in its microstructure and formation of fragments with the zonal shell structure. Thus, nanoparticles in the composition of (BeO + TiO_2_) ceramics contribute to restraining the growth of micron crystals of BeO and TiO_2_ under the action of surface and bulk diffusion processes. The introduction of TiO_2_ nanoparticles contributes to the densification of the ceramic structure due to the reduction of intergranular boundaries between TiO_2_ and BeO, and it also revealed the existence of TiO_2_ solid phase and liquid solution spreading at a certain temperature on the surface of BeO microcrystals.

The addition of TiO_2_ nanoparticles to the composition of the studied ceramics leads to changes in mass transfer mechanisms for micro- and nanoparticles, which allows for synthesizing a material with increased density, hardness, and mechanical strength.

According to the results of studies on the frequency dependence of the reflection coefficient on the content of TiO_2_ nanoparticles, it was found that the introduction of nanoparticles in the amount of 1.0–1.5 wt.%, and an increase in the sintering temperature of ceramics, leads to an increase in the ability of the material to absorb electromagnetic radiation in the frequency range of an electric current between 8.2–12.4 GHz.

Using the XRD method, it has been established that at sintering (BeO + TiO_2_) ceramics with nanoparticles addition in a vacuum and a weakly reducing medium CO, in the furnace atmosphere conditions of oxygen deficiency are created, at which the reaction 2TiO_2_ + C → Ti_2_O_3_ + CO is preferable. In turn, when Ti_2_O_3_ is fused with TiO_2_ under the influence of temperature it forms a phase Ti_3_O_5_, which is a double oxide of tri- and tetravalent titanium $Ti_{2}^{\mathrm{III}}O_{3}+Ti^{\mathrm{IV}}O_{2}\to Ti_{3}^{10/3}O_{5}$. Thus, the increase in sintering temperature and chemical activity of TiO_2_ nanoparticles leads to an increase in the content of the Ti_3_O_5_ phase, and consequently, to an increase in the ability of the material to absorb electromagnetic radiation. Thus, quantitative change in the ratio of BeO and titanium-containing phases, as well as regulation of the particle size distribution and degree of non-stoichiometry of titanium oxides, allows us to effectively control the electrophysical properties of synthesized ceramics.

## Figures and Tables

**Figure 1 materials-16-06507-f001:**
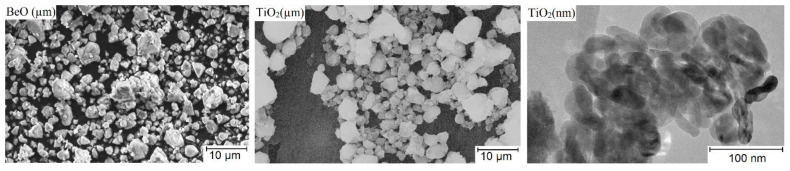
Electron image of beryllium oxide powder microparticles, titanium dioxide microparticles and nanoparticles.

**Figure 2 materials-16-06507-f002:**
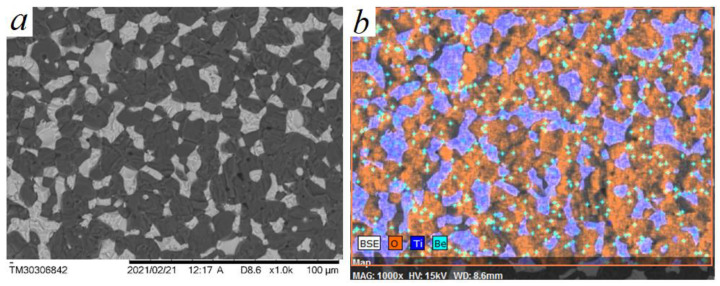
Microstructure of the ceramic sample of composition: BeO + 30%TiO_2_, obtained from micropowders of BeO and TiO_2_, sintering temperature T = 1530 °C, P0 composition sample. (**a**) SEM image of the microstructure of (BeO + TiO_2_) ceramics; (**b**) results of microanalysis of structural components of (BeO + TiO_2_) ceramics.

**Figure 3 materials-16-06507-f003:**
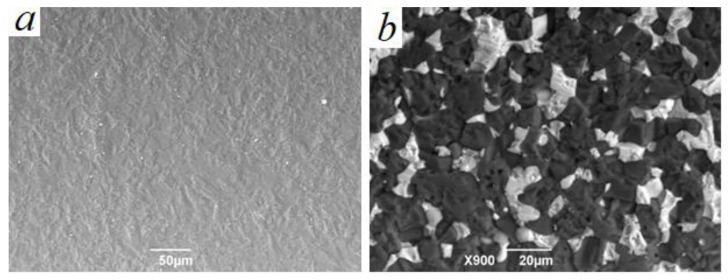
Microstructure of pure BeO ceramics and ceramics consisting of BeO and TiO_2_ micropowders, P0 composition sample: (**a**) SEM image of microstructure of BeO ceramics without impurities; (**b**) SEM image of microstructure of serial (BeO + TiO_2_) ceramics consisting of micropowders.

**Figure 4 materials-16-06507-f004:**
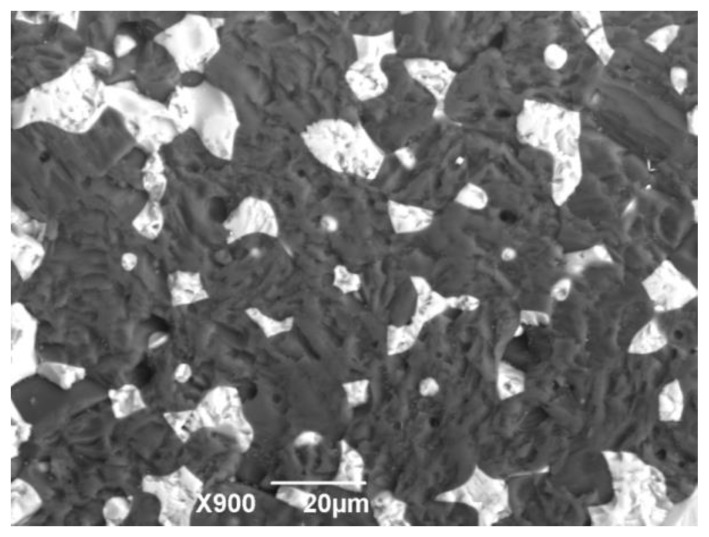
SEM image of microstructure of ceramics with nanoparticles addition of 1.0 wt.%, sintering temperature T = 1660 °C, P2 composition sample.

**Figure 5 materials-16-06507-f005:**
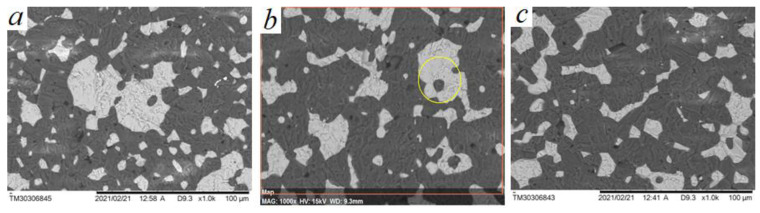
SEM image of the microstructure of ceramics with TiO_2_ nanoparticles, sintering temperature T = 1660 °C: (**a**) P1; (**b**) P2; (**c**) P3.

**Figure 6 materials-16-06507-f006:**
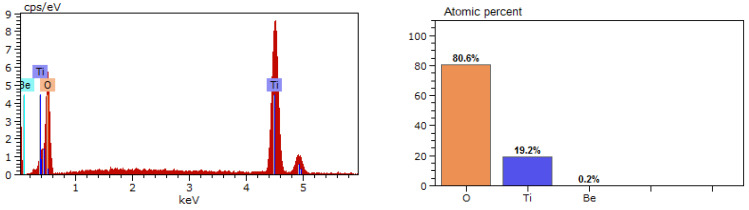
Chemical microanalysis of the element of the zone shell structure.

**Figure 7 materials-16-06507-f007:**
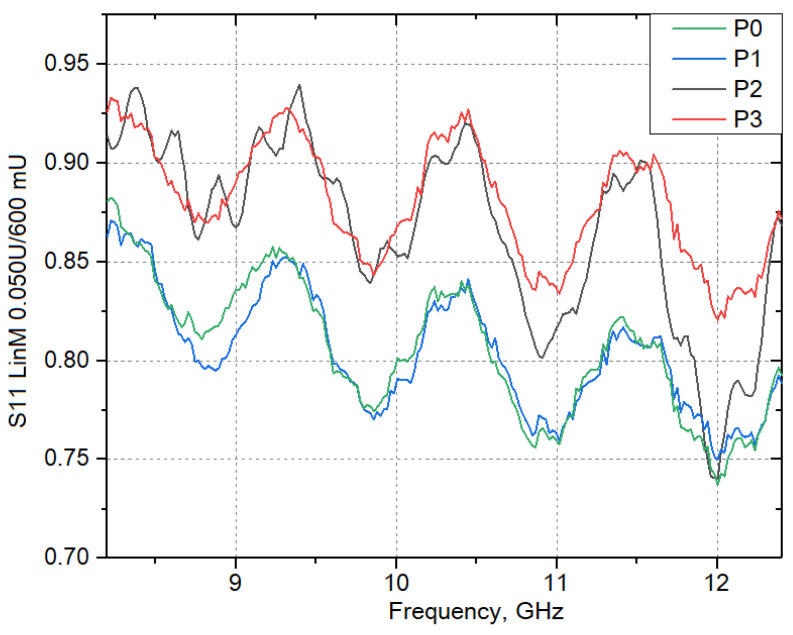
Variation of reflection coefficient (S11/S22 Lin Mag) in the frequency range of 8.2–12.4 GHz, depending on the concentration of TiO_2_ nanoparticles.

**Figure 8 materials-16-06507-f008:**
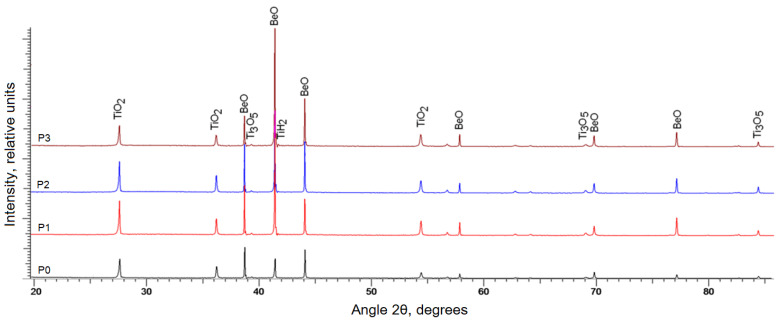
X-ray diffractograms of the studied ceramics. P0—serial sample, P1–P3—samples sintered at 1660 °C with different concentration of TiO_2_ nanoparticles (0.5–1.5) wt.%.

**Table 1 materials-16-06507-t001:** Compositions of the studied samples.

No. of Batch	Component Content, wt.%		
	BeO	TiO_2_ (µm)	TiO_2_ (Nano)
P0	70	30	-
P1	70	29.5	0.5
P2	70	29.0	1.0
P3	70	28.5	1.5

**Table 2 materials-16-06507-t002:** Calculation of Gibbs energy change, which determines the possibility of spontaneous reaction at T = 1660 °C.

Chemical Reaction	ΔG, kJ/mol
TiO_2_ + C → TiO + CO	ΔG_T_ < 0
2TiO_2_ + C → Ti_2_O_3_ + CO	
3TiO_2_ + H_2_ →Ti_3_O_5_ + H_2_O	
TiO_2_ + CH → CO_2_ + TiH	

**Table 3 materials-16-06507-t003:** Physical and mechanical properties of synthesized ceramics.

No. of Batch	Mechanical Strength, MPa	Microhardness, GPa	Density, g/cm^3^	Total Porosity, %
P0	280	9.3	3.22	7.1
P1	295	9.7	3.23	5.9
P2	300	10.0	3.24	5.3
P3	300	10.0	3.24	5.3

**Table 4 materials-16-06507-t004:** Properties of ceramics modified with TiO_2_ nanoparticles in comparison with the sample consisting only of micropowders.

Parameter	$BeO+30\%TiO_{2}^{\mu m}$	$BeO+28.5\%TiO_{2}^{\mu m}+1.5\%TiO_{2}^{nano}$
Number of grains with the size of 1–5 microns on the area of 0.137 mm^2^, pcs.	334	1257
Apparent density, g/cm^3^	3.2	3.24
Hardness, GPa	9.3	9.6
Mechanical strength, MPa	280	300

**Table 5 materials-16-06507-t005:** Phase composition and lattice parameters of synthesized ceramics in comparison with the serial sample.

Phase	Type of Structure	P0, Content%	P1, Content%	P2, Content%	P3, Content%
TiO_2_ (rutile)	Tetragonal	49.0	41.3	39.8	40.4
BeO	Hexagonal	37.1	32.1	30.1	30.0
TiH_2_	Tetragonal	3.7	6.9	7.7	7.2
Ti_3_O_5_	Orthorhombic	10.2	19.7	22.4	22.4

## Data Availability

Not applicable.

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
