# Peer review of "Synthesis and Investigation of Properties of Beryllium Ceramics Modified with Titanium Dioxide Nanoparticles"

_materials, 2023, doi:10.3390/ma16196507_

Round 1
Reviewer 1 Report
The article can comprehensively improve the properties of BeO-TiO2 composites by adding TiO2 nanoparticles, the density, mechanical strength, hardness, and electromagnetic absorption are improved, which helps to inhibit the growth of micrometer crystals of beryllium oxide and increase the sintering temperature from 1530 ℃ to 1660 ℃, the study is meaningful, but some questions need to be answered by the authors:
1. The introduction spends a lot of time on the existing research on the (BeO-TiO2) system, but it does not seem to point out the shortcomings of the current stage of the research, as well as the article seems to change the content of TiO2 nano-particles in TiO2 only based on the research on BeO + 30% TiO2, and what is the basis for doing this?
2. Temperature units in phase diagrams should preferably be written in full.
3. The content of TiO2 nanoparticles in the paper is only added up to 1.5 wt%, at which point the performance is excellent, did the authors try a higher content scenario?
4. paragraph 7.8.9 of the text describes the effect of a co-reducing environment on the conductivity and electromagnetic radiation absorption capacity of a material, but later in the text there is no comparison of the content of CO, is the situation here to be described.
Author Response
|
Thank you for giving me the opportunity to submit a revised version of my manuscript. I appreciate the time and effort you spent in providing valuable feedback on my manuscript. Below we provide detailed responses as well as the relevant changes and corrections made. |

Reviewer 2 Report
the idea of introducing nanocrystalline TiO2 to the mixture of BeO - TiO2 (microstructure) is not new. The authors have published similar paper dealing with this idea (https://doi.org/10.2478/msp-2022-0003)
Please, precise the relevance of the present results compared to the previous.
Can the authors clarify how they deduce the presence of the Ti3O5 phase without performing the XRD measurements and analysis?
Some editing for the English language is required throughout the manuscript due to severals errors.
Author Response

(The authors gave the same response as above.)

Reviewer 3 Report
New Beryllium ceramics have been synthesised with the addition of micro- and nanoparticles TiO2 by the method of thermoplastic slip casting.
The addition of TiO2 - nanoparticles modify the properties of the final ceramics.
The texture modifications are followed by SEM and EDX and then an increase in the ability of the material to absorb electromagnetic radiation in the frequency range of electric current 8.2 - 12.4 GHz was demonstrated.
Unfortunately, the manuscript is not well organised and is difficult to read, there are many processes simply described without any experimental evidence.
The use of other characterisation techniques as diffraction or TEM should give more information on the actual composition of the samples.
Before the paper can be accepted for publication, the manuscript should be revised to clearly establish the scientific reason, the methodology applied to characterise the new ceramics, the structural differences between the four samples and clearly explain the origin of the differences in absorption properties.
The quality of the text must be improved, particularly the style.
Author Response
Thank you for your attention to this manuscript. In the attached files, we provide detailed responses and the relevant changes and corrections are addressed.

Round 2
Reviewer 2 Report
The paper is largely improved after the revision.
It can be accepted for publicatiuon in Materials.
Reviewer 3 Report
The authors have responded to the referee's queries and requests and the manuscript has been significantly improved. It can be accepted for publication.